# Unfavourable sedentary and physical activity behaviour before and after retirement: a population-based cohort study

Nienke ter Hoeve,[1,2] Maria Ekblom,[3,4] Maria R Galanti [ID],[5] Yvonne Forsell,[5]
Carla F J Nooijen [ID] [3,6]

[1]Capri Cardiac Rehabilitation, Rotterdam, The Netherlands
[2]Department of Rehabilitation Medicine, Erasmus University Medical Center, Rotterdam, The Netherlands
[3]The Swedish School of Sport and Health Sciences (GIH), Stockholm, Sweden
[4]The Department of Neuroscience, Karolinska Institutet, Stockholm, Sweden
[5]The Department of Global Public Health, Karolinska Institutet, and Centre for Epidemiology and Community Medicine, Stockholm, Sweden
[6]The Department of Global Public Health, Karolinska Institutet, Stockholm, Sweden

**Correspondence to**
Dr Carla F J Nooijen;
carla.nooijen@gih.se

## ABSTRACT

**Background** During transition to retirement there is often a rearrangement of daily life which might provide a key opportunity for interventions to promote a non-sedentary and active lifestyle. To be able to design effective interventions, it is essential to know which sedentary and physical behaviour domains (eg, at home or during leisure time) have potential to facilitate healthy ageing during the retirement transition.
**Objective** To determine whether unfavourable sedentary and physical activity behaviour before retirement predict unfavourable sedentary and physical activity behaviour after retirement.
**Design** Population-based cohort.
**Setting and participants** Adults (n=3272) employed in 2010 but retired in 2014.
**Methods** Self-reported preretirement job activity, sedentary leisure time, physical activity at home, and walking-cycling and exercise were assessed as predictors for unfavourable sedentary and physical activity behaviours after retirement using logistic regression. Unfavourable behaviours were defined based on the respective median of the cohort distribution. Furthermore, the OR for having multiple unfavourable behaviours after retirement was determined, based on the amount of unfavourable behaviours before retirement. All models were adjusted for gender and education.
**Results** Unfavourable preretirement physical activity and sedentary behaviour at home or during leisure time were the strongest predictors of the same behaviours after retirement. Unfavourable job activity did not predict physical activity but did predict unfavourable sedentary behaviour after retirement (OR=1.66, 95% CI 1.41 to 1.96). Unfavourable exercise behaviour before retirement predicted unfavourable sedentary and physical activity after retirement in all domains. With all behaviours being unfavourable before retirement, the OR of having at least three unfavourable behaviours after retirement was 36.7 (95% CI 16.8 to 80.5).
**Conclusions** Adults with a higher number of unfavourable preretirement physical activity and sedentary behaviours are likely to carry these unfavourable behaviours into retirement age. Interventions should target those with more unfavourable preretirement physical activity and sedentary behaviours before retirement, and those interventions focusing on exercise might have greatest potential.

## Strengths and limitations of this study

► The study described a large longitudinal cohort investigating changes in sedentary behaviour and physical activity during retirement transition.
► The used instrument (Physical Activity Questionnaire) takes into account both physical activity and sedentary behaviour in different domains (eg, at work, during leisure time).
► This study provides valuable knowledge for public health researchers and policymakers, indicating that interventions preferably focusing on exercise should target individuals with unfavourable physical activity and sedentary behaviours before retirement.
► We did not have information on the exact time since retirement.

## INTRODUCTION

An ageing population and increasing life expectancy result in a growing number of adults spending a long time in retirement.[1] Sedentary behaviour and physical activity are two related and independent predictors of healthy ageing.[2] More sedentary time and less physical activity are related to increased risk of diabetes, cardiovascular disease and all-cause mortality.[3–7] Studies in older adults have additionally shown that unfavourable sedentary and physical activity patterns increase the risk of functional limitations in the performance of activities of daily life such as walking and performing house chores.[8]

Elderly spend 65%–80% of their waking hours sedentary and only a minority meet physical activity recommendations.[9 10] During transition to retirement, daily life often undergoes a rearrangement which might provide a key opportunity for interventions to stimulate a non-sedentary and active lifestyle. Lack of time is a frequently reported barrier to physical activity, which might not exist any longer after retirement.[11]

Results of previous studies investigating changes in sedentary and physical activity behaviour during transition from working life to retirement identified varying patterns of change dependent on study methodology. Most studies used single item questions and did not focus on different domains simultaneously (eg, at work, during leisure time).[12 13] With regard to sedentary behaviour, both decreases in total sitting time and an increase in sedentary leisure activities such as watching TV have been reported.[12 14 15] With regard to physical activity, decreases were mainly seen in occupational physical activity and transport physical activity,[14 16] while increases were reported in time spent in leisure time physical activity and walking.[15 17–19] The domain in which the sedentary and physical activity behaviour is performed is important since determinants and health effects may be different.[20 21] To be able to design effective interventions, it is essential to know which sedentary and physical behaviour domains have potential to facilitate healthy ageing during the retirement transition.

A previous study showed that changing from an active to a less active occupation was compensated by exercising more during leisure time, and the other way around.[22] It is therefore conceivable that persons who retire from a physically active occupation might compensate by being more active during leisure time activities after retirement. In line with this hypothesis, it has been suggested that persons with a high occupational sitting time and low levels of physical activity before retirement are at risk for adverse sedentary behaviour outcomes after retirement.[14] Furthermore, a large cohort study found a relation between increased walking time and decreasing sedentary leisure time after retirement.[19] On the contrary, in another cohort study, no relations were found between changes in time spent in different physical activity and sedentary behaviours during retirement.[15] To get deeper insight into behaviours related to sedentary behaviour and physical activity after retirement, more large-scale research that simultaneously investigates the relation between sedentary and physical activity behaviour in different domains both before and after retirement is warranted.

Our aim was to determine which sedentary and physical activity behaviours are predictors of unfavourable sedentary and physical activity behaviour after retirement. We analysed whether sedentary and physical activity behaviour after retirement can be predicted by preretirement sedentary behaviour and physical activity in different domains (at work, at home or during leisure time) in a large population-based cohort including persons who recently retired. We hypothesised that having a more sedentary occupation, unfavourable leisure physical activity or unfavourable leisure sedentary behaviour before retirement predicts higher levels of sedentary behaviour and lower levels of physical activity behaviour after retirement. This study will add with valuable knowledge for public health and policymakers on who to target and which physical activity and sedentary

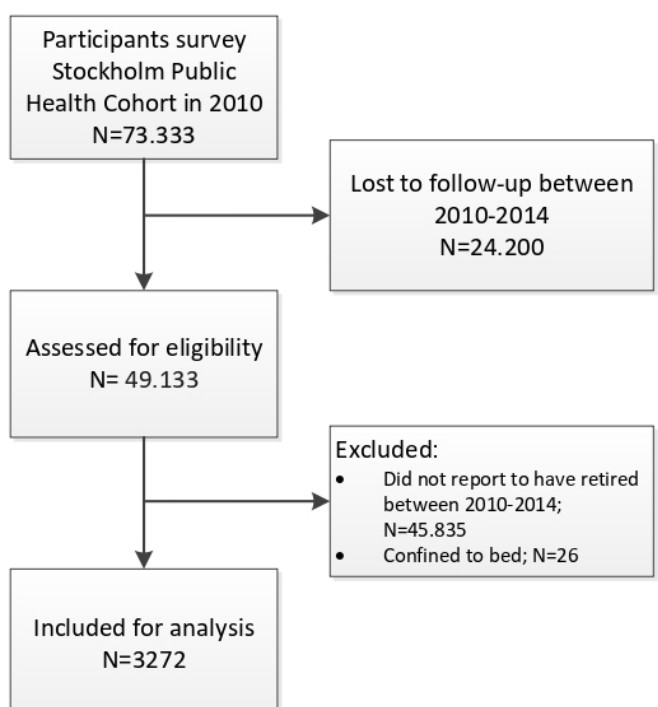

**Figure 1** Flow chart of participants.

behaviour to target to improve healthy ageing during the retirement transition.

## METHODS
### Study population
We used data from the Stockholm Public Health Cohort, a large population-based cohort in Stockholm County.[23] Population samples were randomly selected from Statistics Sweden's Register of the total population, after stratification according to residential municipality. Every 4 years, participants from all three samples completed similar questionnaire-based surveys on a range of demographic and health variables. Register data from Statistics Sweden have been linked to the self-reported information.

Of the participants who completed the survey in 2010, 67% completed the follow-up survey in 2014.[23] Participants who reported information on physical activity and sedentary behaviour in both 2010 (baseline) and 2014 (follow-up) were assessed for eligibility (n=49 133). Mean age of the sample assessed for eligibility was 54 years (SD=16), 57% were women and 48% highly educated. Participants were excluded if they were confined to bed on both occasions (n=26). In the current study, those who were employed or self-employed in 2010 but full time retired in 2014 were included, resulting in an analytical sample of 3272 participants (see figure 1).

### Physical activity and sedentary behaviour assessment
Physical activity and sedentary behaviour were assessed with the Physical Activity Questionnaire (PAQ), which has shown to be valid for classification into physically active or sedentary.[24 25] For all questions, participants were

ter Hoeve N, *et al. BMJ Open* 2020;**10**:e037659. doi:10.1136/bmjopen-2020-037659

requested to answer on their average behaviour during the past 12 months, considering the week variability and seasonality.

Physical activity and sedentary behaviour in the following domains were assessed with this questionnaire:

▶ *Job activity*
  With six answer categories: 'mainly sedentary', 'sitting approximately half of the time', 'mainly standing', 'walking mostly, lifting, carrying a little', 'walking mainly, lifting, carrying a lot' or 'heavy physical work'.

▶ *Sedentary leisure time*
  This includes leisure sitting time, for example, watching TV or reading. There were seven response categories ranging from less than 1 hour/day to more than 6 hours/day.

▶ *Physical activity at home*
  This included physical activity during home, household and gardening tasks. There were six response categories ranging from less than 1 hour/day to more than 5 hours/day.

▶ *Walking and cycling activity*
  With six response categories for answering combined walking and cycling activity, ranging from hardly ever to more than 2 hours/day.

▶ *Exercise*
  A question was asked about hours of exercise per week, excluding daily walking and cycling. Participants chose one of seven options ranging from hardly ever to more than 5 hours/week.

## Data analyses

The cohort median for all physical activity and sedentary behaviour variables was determined and used to dichotomise behaviours into more favourable versus unfavourable. The medians were determined separately at baseline and at follow-up surveys, since occupational activity only applied to baseline and therefore the distribution of physical activity and sedentary behaviour domains was naturally different at follow-up after retirement.[26] Unfavourable job activity at work (only baseline) was defined as mainly sedentary or sitting approximately half of the time. Unfavourable sedentary behaviour during leisure time at baseline was defined as 2–3 hours/day or more, and at follow-up as 3–4 hours/day or more. Physical activity at home was defined as unfavourable when less than 1 hour/day at baseline and less than 1–2 hours/day at follow-up. For both baseline and follow-up, walking and cycling activity was defined as unfavourable when performed for less than 20 min/day, and exercise when accumulating less than 1 hour/week.

Binary logistic regression analyses were conducted using general linear models in order to derive ORs of unfavourable behaviour after retirement for: (1) sedentary time, (2) physical activity at home, (3) walking/cycling, and (4) exercise. Predictors were unfavourable behaviours before retirement related to: (A) job activity, (B) sedentary leisure time, (C) physical activity at home, (D) walking/cycling, and (E) exercise.

A similar additional logistic regression analysis was conducted for all behaviours considered together, that is, estimating the amount of unfavourable behaviours after retirement based on the amount of unfavourable behaviours before retirement. To obtain a dichotomous variable, the median number of unfavourable behaviours after retirement was used, that is, three or more unfavourable behaviours.

Education was used as an indicator of socioeconomic position, and all models were adjusted for education and gender

We checked for multicollinearity by assessing both correlations between the predictors and with variance inflation factor statistics and concluded that there were no indications of collinearity. All statistical analyses were performed in IBM SPSS Statistics V.24 (IBM).

## RESULTS

Mean age of the analytical sample at baseline was 63 years (SD=2), 55% were women and 43% highly educated. The associations between preretirement and after-retirement behaviours are shown in table 1.

The results can be summarised as:

▶ Unfavourable preretirement physical activity and sedentary behaviour at home or during leisure time were the strongest predictors of the same behaviour after retirement.

▶ Unfavourable job physical activity predicted unfavourable sedentary time after retirement, but not any other unfavourable physical activity behaviour.

▶ Unfavourable exercise behaviour before retirement predicted all unfavourable sedentary and physical activity behaviours after retirement.

▶ Unfavourable sedentary leisure time predicted all unfavourable behaviours, except for walking/cycling.

The relation between multiple unfavourable behaviours before and after retirement is shown descriptively in table 2. Among participants who presented with all unfavourable behaviours at baseline the adjusted OR of having three or more unfavourable behaviours after retirement (vs 2 or less) was 36.7 (95% CI 16.8 to 80.5). The corresponding OR for four unfavourable behaviours before retirement was 14.3 (95% CI 7.1 to 29.0); 6.6 (95% CI 3.3 to 13.2) for three unfavourable behaviours, 3.2 (95% CI 1.6 to 6.5) for two unfavourable behaviours and 1.7 (95% CI 0.8 to 3.6) for one unfavourable behaviour.

## DISCUSSION

This longitudinal study increases the understanding of sedentary and physical activity behaviour during the retirement transition. We found that unfavourable preretirement physical activity and sedentary behaviour at home or during leisure time were the strongest predictors of the same behaviour after retirement. Preretirement job activity did not predict low levels of physical activity but did predict sedentary behaviour after retirement. Furthermore, less

**Table 1** Baseline characteristics and predictors of more unfavourable sedentary and physical activity behaviour after retirement

| | Sedentary leisure time after retirement | | | |
| | Unfavourable after retirement | Favourable after retirement | | |
| | n=1441 | n=1805 | | |
| **Predictors of unfavourable leisure time sedentary behaviour after retirement*** | % (n) | % (n) | OR | 95% CI |
| Unfavourable job activity before retirement | 70 (988) | 59 (1042) | **1.66** | **1.41 to 1.96** |
| Unfavourable sedentary leisure time before retirement | 66 (933) | 33 (588) | **4.19** | **3.59 to 4.89** |
| Unfavourable activity at home before retirement | 36 (512) | 29 (522) | **1.32** | **1.12 to 1.56** |
| Unfavourable walking/cycling before retirement | 72 (1029) | 67 (1194) | 1.17 | 0.99 to 1.38 |
| Unfavourable exercise before retirement | 48 (678) | 40 (709) | **1.29** | **1.10 to 1.50** |
| | Physical activity at home after retirement | | | |
| | Unfavourable after retirement | Favourable after retirement | | |
| | n=1928 | n=1302 | | |
| **Predictors of unfavourable home physical activity after retirement*** | % (n) | % (n) | OR | 95% CI |
| Unfavourable job activity before retirement | 66 (1252) | 60 (769) | 1.16 | 0.99 to 1.37 |
| Unfavourable sedentary leisure time before retirement | 50 (957) | 44 (562) | **1.35** | **1.16 to 1.58** |
| Unfavourable activity at home before retirement | 44 (837) | 15 (188) | **3.92** | **3.26 to 4.72** |
| Unfavourable walking/cycling before retirement | 71 (1359) | 66 (850) | 1.06 | 0.90 to 1.25 |
| Unfavourable exercise before retirement | 46 (878) | 39 (503) | **1.23** | **1.05 to 1.44** |
| | Walking/cycling after retirement | | | |
| | Unfavourable after retirement | Favourable after retirement | | |
| | n=1923 | n=1313 | | |
| **Predictors of unfavourable walking/cycling after retirement*** | % (n) | % (n) | OR | 95% CI |
| Unfavourable job activity before retirement | 66 (1240) | 61 (787) | 1.07 | 0.91 to 1.26 |
| Unfavourable sedentary leisure time before retirement | 47 (902) | 48 (620) | 0.99 | 0.84 to 1.15 |
| Unfavourable activity at home before retirement | 34 (651) | 29 (374) | 1.04 | 0.88 to 1.23 |
| Unfavourable walking/cycling before retirement | 81 (1537) | 53 (681) | **3.74** | **3.18 to 4.41** |
| Unfavourable exercise before retirement | 48 (911) | 37 (474) | **1.43** | **1.22 to 1.66** |
| | Exercise after retirement | | | |
| | Unfavourable after retirement | Favourable after retirement | | |
| | n=1414 | n=1833 | | |
| **Predictors of unfavourable exercise after retirement*** | % (n) | % (n) | OR | 95% CI |
| Unfavourable job activity before retirement | 65 (896) | 63 (1134) | 1.07 | 0.91 to 1.26 |
| Unfavourable sedentary leisure time before retirement | 52 (731) | 44 (791) | **1.36** | **1.17 to 1.59** |
| Unfavourable activity at home before retirement | 35 (483) | 30 (547) | 1.05 | 0.89 to 1.24 |
| Unfavourable walking/cycling before retirement | 73 (1021) | 67 (1205) | 1.15 | 0.97 to 1.36 |
| Unfavourable exercise before retirement | 64 (900) | 26 (478) | **4.92** | **4.21 to 5.75** |

*Adjusted for gender and education.

than 1 hour exercise per week before retirement predicted all of the unfavourable behaviours after retirement. Unfavourable sedentary leisure time before retirement predicted both sedentary time and most physical activity behaviours after retirement. Moreover, the higher the number of unfavourable behaviours before retirement, the more likely it

ter Hoeve N, *et al. BMJ Open* 2020;**10**:e037659. doi:10.1136/bmjopen-2020-037659

**Table 2** Multiple unfavourable behaviours before and after retirement

| | | Number of unfavourable behaviours after retirement (n=3092) | | | | |
|---|---|---|---|---|---|---|
| | % (n) | All 10 (309) | 3 24 (755) | 2 35 (1087) | 1 24 (732) | None 7 (209) |
| Number of unfavourable behaviours before retirement | All | 43 (68) | 35 (55) | 18 (28) | 4 (6) | 0 (0) |
| | 4 | 22 (109) | 36 (182) | 32 (160) | 10 (50) | 1 (4) |
| | 3 | 9 (83) | 30 (282) | 39 (369) | 19 (183) | 3 (32) |
| | 2 | 4 (40) | 19 (170) | 37 (340) | 31 (286) | 8 (74) |
| | 1 | 2 (9) | 12 (56) | 35 (163) | 35 (163) | 16 (74) |
| | None | 0 (0) | 9 (10) | 26 (27) | 42 (44) | 24 (25) |

was that a person had multiple (at least three) unfavourable behaviours after retirement.

Contrary to our hypothesis, preretirement job activity did not seem to be related to postretirement physical activity in this study, that is, there was no hint that persons retiring from physically active jobs compensated the loss of this activity by increasing leisure time physical activity. A previous study showed that people changing jobs do compensate loss of job activity in their leisure time.[22] Retirement means a change to no job activity at all, potentially explaining why this compensation does not apply. Nevertheless, job activity did seem to predict sedentary behaviour after retirement in the anticipated direction. These results are in line with a previous study where decreases in occupational sedentary behaviour were compensated by increasing sitting time outside working hours.[27]

A systematic review[12] suggested that preretirement physical activity and sedentary behaviour outside work may be stronger predictors of behaviour after retirement compared with job activity. This hypothesis is supported by our findings that unfavourable preretirement physical activity and sedentary behaviour at home or during leisure time were the strongest predictors of the same unfavourable behaviours after retirement. These results confirm that behaviours tend to be rather stable over the life course,[28] and this is probably true even after a major life change such as retirement.

We found that unfavourable exercise behaviour before retirement seems to influence practically all unfavourable physical activity and sedentary behaviours after retirement. It is possible that regular exercise results in a higher physical fitness level, subsequently in a lower strain when performing physical activities, making it easier to maintain a favourable level of both physical activity and sedentary behaviour even at older age. This outcome is in line with a previous study that found that persons with low levels of physical activity before retirement are at risk for unfavourable sedentary behaviour after retirement.[14] In addition to exercise behaviour, unfavourable sedentary leisure time before retirement predicted both sedentary time and most physical activity behaviours after retirement. These outcomes suggest that interventions promoting exercise behaviour and limiting sedentary time before retirement

may potentially prevent unfavourable physical activity and sedentary behaviour also after retirement.

Older adults often have unfavourable physical activity and sedentary behaviour[9 10] which negatively impacts healthy ageing.[3–8] Our results imply that adults with multiple unfavourable preretirement behaviours are at higher risk to hold this profile after retirement. This group should be thought of as a priority for preventive interventions targeting physical activity and sedentary behaviour. Despite the paucity of studies of interventions during the retirement transition, a review concluded that different types of counselling programmes, such as group sessions, individual training sessions, in-home exercise programmes or e-health programmes, can lead to positive effects in ageing adults.[29] Our results carry a decisive suggestion that future studies should evaluate the effectiveness of preretirement exercise and sedentary behaviour interventions on overall postretirement physical activity and sedentary behaviour.

When interpreting the results of the study it should be realised that we used the sample medians to distinguish between favourable and unfavourable behaviours. Since there is no evidence for cut-off points for domain-specific physical activity and sedentary behaviour, it is unclear whether the medians reflect a true border between favourable and unfavourable. The question on job activity was a combination of sedentary behaviour and physical activity and our definition of favourable was being more active at work. However, a recent review indicates that high levels of job activity might have detrimental health consequences and therefore not necessarily be favourable.[30] Furthermore, we did not separate different leisure time sedentary activities (such as computer use, TV, reading, and so on) while certain leisure sitting time might be more likely to be combined with other unfavourable health behaviours such as drinking alcohol and eating snacks.[31] In addition, we did also not separate between walking and cycling and between different exercise modalities even though we are aware that health effects might depend on the intensity and type of exercise performed.

### Strengths and limitations

The used instrument (PAQ) takes into account both physical activity and sedentary behaviour in different domains and was used to measure these behaviours both

before and after retirement, which is a major strength of our study. Nevertheless, as discussed above, the instrument does not differentiate between vocational physical activity and sedentary behaviour, between different types of leisure sedentary activities and between types of exercise performed. There were also some limitations. First, all participants who retired in a time frame of 4 years were included in this study. There was no information available on the exact time of retirement, while this could have influenced behavioural adjustments. Second, we had no information on possible confounders of outcomes, such as comorbidities. Third, we included persons who completed two surveys, which might have led to a selection bias. A relatively large proportion of the included sample was highly educated, which is known to be related to more favourable exercise patterns and sedentary behaviour.[32–34] Furthermore, a decrease in income after retirement might have potentially influenced retirement behaviours including physical activity and sedentary behaviour. Last, we cannot rule out misclassification of sedentary behaviour and physical activity since it is known that self-report measures often demonstrate restricted validity and reliability.[35]

## CONCLUSIONS

This study contributes to a deeper understanding of unhealthy ageing with novel insights on unfavourable sedentary and physical activity behaviour after retirement. Despite the major life event of retirement, preretirement unfavourable behaviours seem likely to be carried into retirement age. Likewise, those with multiple unfavourable preretirement behaviours seem at a higher risk to hold the same unfavourable profile after retirement. There was no indication that persons retiring from physically active jobs compensated the loss of this activity by increasing leisure time physical activity. Interventions should target those with more unfavourable preretirement physical activity and sedentary behaviours before retirement, and those interventions focusing on exercise might have greatest potential.

**Contributors** NtH and CFJN planned the study. NtH, ME, MRG, YF and CFJN contributed to design and methodology of the manuscript. CFJN was responsible for data analysis and data interpretation. NtH, ME, MRG and YF contributed to interpreting the data. NtH was responsible for drafting the manuscript and revision of the manuscript after review. All authors read and approved the final manuscript.

**Funding** The work of CFJN was supported by the Swedish Research Council for Health, Working Life and Welfare grant number (FORTE (2017-01385)).

**Competing interests** None declared.

**Patient and public involvement** Patients and/or the public were not involved in the design, or conduct, or reporting, or dissemination plans of this research.

**Patient consent for publication** Not required.

**Ethics approval** The present study was approved by the Stockholm Regional Ethical Review Board (case number: 2016/749-32).

**Provenance and peer review** Not commissioned; externally peer reviewed.

**Data availability statement** Data may be obtained from a third party and are not publicly available. Applications for access to data can be send to the

Stockholm County Council, https://www.folkhalsoguiden.se/halsa-stockholm/halsa-stockholm---for-forskare/

**ORCID iDs**
Maria R Galanti http://orcid.org/0000-0002-7805-280X
Carla F J Nooijen http://orcid.org/0000-0003-0146-9292

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
