## [Reviewer comments · BMJ Open]

ARTICLE DETAILS

TITLE (PROVISIONAL)	Unfavourable sedentary and physical activity behaviour before and after retirement. A population-based cohort study
AUTHORS	ter Hoeve, Nienke; Ekblom, Maria; Galanti, Maria; Forsell, Yvonne; Nooijen, Carla

VERSION 1 – REVIEW

REVIEWER	Jean-Michel Oppert Sorbonne university, Paris, France
REVIEW RETURNED	23-Feb-2020

GENERAL COMMENTS	General comments Authors assessed whether self-reported pre-retirement job activity, sedentary leisure time, physical activity at home, walking-cycling and exercise were predictors for unfavourable sedentary and physical activity behaviours after retirement. Data came from a population-based study and included 3,272 individuals followed between 2010 and 2014 before and after they retired. Results show that adults with a higher number of unfavourable pre-retirement physical activity and sedentary behaviours were likely to carry these unfavourable behaviours into retirement age. Job activity was not a predictor. It is concluded that physical activity interventions before retirement may have the potential to prevent unhealthy behaviours during the retirement period. This is an interesting topic in public health considering the growing proportion of retired individuals in the aging population of most industrialised countries. A major strength of the study is the large sample of retirees included from a population-based cohort and the variety of behaviours assessed. There are however a number of issues that should be dealt with: i) The literature cited misses a number of relevant papers on the topic. Especially, some of those that have specifically tried to address the issue of the changes with retirement in the different domains of physical activity and sedentary behaviour (see specific comments). ii) The instrument used (PAQ questionnaire) has advantages over many other used in the literature that are even more simplistic, however it does not differentiate the different types of leisure sedentary activities, and this is a limitation that should be noted. It does not differentiate either between walking or cycling, an issue of current major interest regarding active transport. The question on exercise is rather crude as there is no knowledge that can be obtained on the type of physical activity performed by subjects. iii) A major issue is the decision taken to divide the data by the median value to define favourable vs. unfavourable behaviours. The authors have to be commended for discussing this and acknowledging it is a major limitation, however it derives directly from the limited type of questions asked by their PAQ which should be mentioned. iiiii) Some
---

	propositions about the results are not accurate (see below specific comments); iiiii) there could be more emphasis on the results on sedentary behaviour, where pre-retirement behaviour seem to predict a number of post-retirement behaviours, although there is no reflection on the fact that only leisure time sedentary behaviour has been assessed and that sedentary behaviour during work might have great importance in the changes from pre- to post-retirement; iiiiii) finally, it could have been of interest, if feasible, to compare these data with those in subjects of about the same age but not retiring during the study period from the same cohort. Specific comments -Introduction, page 4, line 27: the authors judge results from the literature prior to their study as “inconsistent” with a lack of “well-conducted studies”. Some previous papers that have looked at different domains of physical activity as well as sedentary behaviour should be cited here (eg Touvier et al. IJBNPA 2010, Menai et al. Plos One 2014). - Results, page 7, line 39: The contention “Unfavourable exercise behaviour before retirement predicted all unfavourable sedentary and physical activity behaviours after retirement » does not seem to match with the corresponding table where the ORs for leisure sedentary time and exercise seem to be the only significant results. -References: some references on the topic are missing (see General and previous comments).
--	--

REVIEWER	Sari Stenholm University of Turku, Finland
REVIEW RETURNED	06-Mar-2020

GENERAL COMMENTS	General comments This is a relatively well written manuscript that aims to examine whether unfavourable sedentary and physical activity before retirement predict unfavourable sedentary and physical activity behavior after retirement. The main strengths of the manuscript include a large longitudinal cohort and determination of leisure and work-specific domains of sedentary and physical behavior around an important life transition. However, there are several issues related to analyses and interpretation of the results that should be considered to further improve the quality of the manuscript. Abstract: 1. Results, lines 34-36: The sentence “Unfavourable sedentary and physical activity in a certain domain before retirement was the strongest predictor of the same behavior after retirement” is challenging to understand, consider rephrasing it. It is logical in terms of pre-retirement physical activity and sedentary behavior at home or during leisure time, because these domains are present both before and after retirement. But in terms of job activity, it is more complicated because there are no such domain after retirement. The main result is expressed more comprehensively in the discussion: “-- unfavourable pre-retirement physical activity and sedentary behavior at home or during leisure time were the strongest predictors of the same behavior after retirement”. 2. Conclusions, line 49-52: Concluding that “Pre-retirement exercise interventions may have great potential to improve physical activity and sedentary behaviors and thereby facilitate healthy aging” seems not quite justified based
---

on the results of the current study, because effectiveness of interventions were not assessed and no other behaviors were examined. Because the authors determined whether unfavourable sedentary and physical activity behaviors predict unfavourable behaviors after retirement, a more suitable conclusion may be that interventions should be targeted to those with a higher unfavourable pre-retirement physical activity and sedentary behaviors.

Introduction:

3. Lines 27-55:

Authors should strengthen the introduction by better bringing out what is known about the physical activity and sedentary behavior - related predictors of physical activity and sedentary behavior after retirement, because several previous studies have examined changes in physical activity and sedentary behavior by following people before and after retirement. One example of a study examining domains of sedentary behavior across transition to retirement is a longitudinal cohort study of Leskinen et al. (doi: 10.1136/jech-2017-209958). In the study of Leskinen et al. it was shown that highest increase in leisure sedentary time in the retirement transition was among those who had high occupational sitting time and low physical activity level before retirement.

Methods:

4. Lines 34-44:

It is not quite clear how the analytical sample was chosen and why only 3272 participants out of 49133 were included. A flow chart describing the sample formation would be very useful for the reader. Regarding selection bias, authors should provide information on whether the participants who completed the survey both 2010 and 2014 differ at baseline from those who completed the survey only in 2010. Selection should also be addressed in the Discussion section.

5. Lines 34-44:

The retirement types of the participants should be specified: were only those with full-time statutory retirement included? Or were there also disability retirees and part-time retirees? Disability retirees may differ from those transitioning to full-time statutory retirement, because they may have lower level of physical activity and higher level of sedentary behavior across the retirement transition. Compared to full-time retirees, part-time retirees are more likely to maintain their physical activity and sedentary behavior levels in the retirement transition because they still spend some time at work.

6. Lines 52-59 (data analyses):

It is not clear if the authors took into account the intraindividual correlation between repeated measurements in the analyses. Please clarify. Moreover, it seems that the analyses were adjusted only for education and gender. There are many other potential confounding factors, such as health status, BMI, depression, that should be taken into account to properly interpret the independent effect of pre-retirement behaviors on post-retirement behavior. Authors should conduct additional analyses to address this caveat.

Results:

7. Lines 32-33, please consider the first comment.

Discussion:

8. Lines 25-27:

Related to the earlier comment about previous literature, this

	sentence is not true “To our knowledge, this is the largest cohort study that studied a variety of behaviors related to both physical activity and sedentary time in different domains both before and after retirement.” There are also other large studies including several thousand participants who have been repeatedly followed before and after retirement. Please update accordingly. 9. Lines 29-32: Could the authors provide information on what was the mean time that had passed from the actual retirement date when study participants completed the survey in 2014? If the information is available, it should be possible to examine whether the results are affected by the timing of the retirement. It is possible that retirees are motivated to engage in physical activity just after transition to retirement, but the effect does not last years after transition to retirement. 10. The authors should tone down the description of the study strengths in the Discussion. First, it is not evident that “Our methods are unique in studying a large variety of behaviours related to both sedentary behaviour and physical activity in different domains” since the methodology is based on self-reported questions. Second, based on the reported results it is not clear how this conclusion can be drawn: “This study adds with valuable knowledge for public health researchers and policy makers that of all sedentary behaviour and physical activity domains, leisure time exercise seems to have the greatest potential in pre-retirement interventions that aim to facilitate healthy aging.” Please see earlier comment for conclusions in the Abstract section and modify accordingly.
--	---

VERSION 1 – AUTHOR RESPONSE

Response to reviewer-1:

General comments

Q1. The literature cited misses a number of relevant papers on the topic. Especially, some of those that have specifically tried to address the issue of the changes with retirement in the different domains of physical activity and sedentary behaviour (see specific comments).

A1. In line with the suggestions of both reviewer 1 and reviewer 2 we extended the discussion on previous studies both in our introduction and discussion section, with a main focus on studies looking at both physical activity and sedentary behaviour in different domains. Please see the changes made in the introduction and discussion in the marked copy of our manuscript

Q2. The instrument used (PAQ questionnaire) has advantages over many other used in the literature that are even more simplistic, however it does not differentiate the different types of leisure sedentary activities, and this is a limitation that should be noted. It does not differentiate either between walking or cycling, an issue of current major interest regarding active transport. The question on exercise is rather crude as there is no knowledge that can be obtained on the type of physical activity performed by subjects.

A2. We agree with the reviewer that this is a limitation of the questionnaire. We added this limitation both to our discussion and limitation section. Please see lines 266-275 in our marked copy.

Q3. A major issue is the decision taken to divide the data by the median value to define favourable vs.

unfavourable behaviours. The authors have to be commended for discussing this and acknowledging it is a major limitation, however it derives directly from the limited type of questions asked by their PAQ which should be mentioned.

A3. This is indeed an issue that we felt was important to discuss, as it influences the interpretation of our results. This issue derives both from a lack of clear cut-off points specified for domain specific activities and from the limited type of questions asked in the PAQ. In line with your previous comment (Q2) and your current suggestion, we expanded our discussion on this topic. Please see lines 256-275 in our marked copy.

Q4. Some propositions about the results are not accurate (see below specific comments)

A4. We addressed these propositions below.

Q5. There could be more emphasis on the results on sedentary behaviour, where pre-retirement behaviour seem to predict a number of post-retirement behaviours, although there is no reflection on the fact that only leisure time sedentary behaviour has been assessed and that sedentary behaviour during work might have great importance in the changes from pre- to post-retirement;

A5. We extended the results and discussion section on the predictive value of pre-retirement sedentary behaviour. Please see lines 201-202, 216-218 and 235-245 in the marked copy of our manuscript. The question about job activity in the survey included answer options ranging from mainly sedentary to heavy physical work. (lines 143-146) It is thus a combination of sedentary behaviour and physical activity which is a limitation as discussed in lines 273-275.

Q6. Finally, it could have been of interest, if feasible, to compare these data with those in subjects of about the same age but not retiring during the study period from the same cohort.

A6. Although we think it is an interesting idea, we believe this is outside the scope of the current manuscript as it will mean that we will need to change our research questions. Furthermore, a large part of the age group that we selected reached the legal retirement age in the time period that our study was conducted. Therefore, it will be difficult to select an age-matched group that did not retire in the study period and is also comparable with regard to other baseline characteristics.

Specific comments

Q7. Introduction, page 4, line 27: the authors judge results from the literature prior to their study as “inconsistent” with a lack of “well-conducted studies”. Some previous papers that have looked at different domains of physical activity as well as sedentary behaviour should be cited here (eg Touvier et al. IJBNPA 2010, Menai et al. Plos One 2014).

A7. In line with the suggestions of both reviewer 1 and reviewer 2 we updated the discussion of previous studies. Please see the changes made in the introduction in the marked copy of our manuscript.

Q8. Results, page 7, line 39: The contention “Unfavourable exercise behaviour before retirement predicted all unfavourable sedentary and physical activity behaviours after retirement » does not seem to match with the corresponding table where the ORs for leisure sedentary time and exercise seem to be the only significant results.

A8. We think that the reviewer derived this conclusion from looking at the last part of table 1 showing which behaviours were predictive for exercise behaviour post-retirement (which are indeed leisure sedentary time and exercise). Nevertheless, in this conclusion we point out that pre-retirement

exercise did predict all post-retirement behaviours. To prevent this confusion in interpreting our tables, we updated the titles of all our tables to make clear that is concerns post-retirement outcomes.

Q9. References: some references on the topic are missing (see General and previous comments).

A9. In line with previous comments, we updated the discussion of previous studies and as such the reference list. Please see the changes made in the introduction and discussion section in the marked copy of our manuscript.

Response to reviewer-2:

Abstract:

Q1. Results, lines 34-36:

The sentence “Unfavourable sedentary and physical activity in a certain domain before retirement was the strongest predictor of the same behavior after retirement” is challenging to understand, consider rephrasing it. It is logical in terms of pre-retirement physical activity and sedentary behavior at home or during leisure time, because these domains are present both before and after retirement. But in terms of job activity, it is more complicated because there is no such domain after retirement. The main result is expressed more comprehensively in the discussion: “--unfavourable pre-retirement physical activity and sedentary behavior at home or during leisure time were the strongest predictors of the same behavior after retirement”.

A1. We rephrased this sentence in line with this suggestion. See lines 17-18 in the marked copy.

Q2. Conclusions, line 49-52:

Concluding that “Pre-retirement exercise interventions may have great potential to improve physical activity and sedentary behaviors and thereby facilitate healthy aging” seems not quite justified based on the results of the current study, because effectiveness of interventions were not assessed and no other behaviors were examined. Because the authors determined whether unfavourable sedentary and physical activity behaviors predict unfavourable behaviors after retirement, a more suitable conclusion may be that interventions should be targeted to those with a higher unfavourable pre-retirement physical activity and sedentary behaviors.

A2. We agree with the reviewer. We adjusted this conclusion to “Interventions should target those with more unfavourable pre-retirement physical activity and sedentary behaviours pre-retirement, and those interventions focusing on exercise might have greatest potential.” See lines 29-31 in the marked copy. In the same line, we also updated our conclusion in lines 298-300 of the marked copy.

Introduction:

Q3. Lines 27-55:

Authors should strengthen the introduction by better bringing out what is known about the physical activity and sedentary behavior -related predictors of physical activity and sedentary behavior after retirement, because several previous studies have examined changes in physical activity and sedentary behavior by following people before and after retirement. One example of a study examining domains of sedentary behavior across transition to retirement is a longitudinal cohort study of Leskinen et al. (doi: 10.1136/jech-2017-209958). In the study of Leskinen et al. it was shown that highest increase in leisure sedentary time in the retirement transition was among those who had high occupational sitting time and low physical activity level before retirement.

A3. We want to thank the reviewer for pointing out this study. In line with the suggestions of both reviewer 1 and reviewer 2 we updated the discussion of previous studies in our introduction section. Please see the changes made in the introduction of the marked copy of our manuscript.

Methods:

Q4. Lines 34-44:

It is not quite clear how the analytical sample was chosen and why only 3272 participants out of 49133 were included. A flow chart describing the sample formation would be very useful for the reader. Regarding selection bias, authors should provide information on whether the participants who completed the survey both 2010 and 2014 differ at baseline from those who completed the survey only in 2010. Selection should also be addressed in the Discussion section.

A4. We added a flow diagram to further clarify how our analytical sample was chosen (see Figure 1). Regarding the selection bias, data provided to the research team only included those who filled out both the questionnaire in 2010 and 2014 and it is therefore not possible to do additional assessments on how the sample might differ. However, we did notice that a relatively large proportion of the included sample was highly educated, which is known to be related to more favourable exercise patterns and sedentary behaviour. Please see our limitation section lines 281-284.

Q5. Lines 34-44:

The retirement types of the participants should be specified: were only those with full-time statutory retirement included? Or were there also disability retirees and part-time retirees? Disability retirees may differ from those transitioning to full-time statutory retirement, because they may have lower level of physical activity and higher level of sedentary behavior across the retirement transition. Compared to full-time retirees, part-time retirees are more likely to maintain their physical activity and sedentary behavior levels in the retirement transition because they still spend some time at work.

A5. We agree with the reviewer that this is important information for the interpretation of our results. We only included people that retired to a full-time statutory retirement. We added this information to our methods section. See line 133 in the marked copy.

Q6. Lines 52-59 (data analyses):

It is not clear if the authors took into account the intraindividual correlation between repeated measurements in the analyses. Please clarify. Moreover, it seems that the analyses were adjusted only for education and gender. There are many other potential confounding factors, such as health status, BMI, depression, that should be taken into account to properly interpret the independent effect of pre-retirement behaviors on post-retirement behavior. Authors should conduct additional analyses to address this caveat.

A6. As we were assessing the relation between behaviours at two time points, we do not believe that we need to take into account the intra-individual correlation between repeated measurements. We agree with the reviewer that the cited variables may represent potential confounders. However, it is not clear how this hypothetical confounding would act, whether as its status at a certain point in time or as a time-dependent change. Also, it would not be clear whether these variables or their changes would be more logically set as confounders or as mediators, since they may reflect effects of PA as well as its predictors. Therefore, we chose to only correct for a minimum amount of time-independent potential confounders. We wish also to underline that we were not aiming at inferring a causal relation between pre- and post- retirement behaviours, but at identifying predictors, i.e. trajectories that may benefit of early interventions.

Results:

Q7. Lines 32-33, please consider the first comment.

A7. We adjusted this sentence in line with the first comment. See lines 193-194 in the marked copy.

Discussion:

Q8. Lines 25-27:

Related to the earlier comment about previous literature, this sentence is not true “To our knowledge, this is the largest cohort study that studied a variety of behaviors related to both physical activity and sedentary time in different domains both before and after retirement.” There are also other large studies including several thousand participants who have been repeatedly followed before and after retirement. Please update accordingly.

A8. We agree with the reviewer that we made this argument too strong and have removed this sentence from our manuscript. Furthermore, we updated the discussion of previous studies in our introduction and discussion section. Please see the changes made in these sections in the marked copy of our manuscript.

Q9. Lines 29-32:

Could the authors provide information on what was the mean time that had passed from the actual retirement date when study participants completed the survey in 2014? If the information is available, it should be possible to examine whether the results are affected by the timing of the retirement. It is possible that retirees are motivated to engage in physical activity just after transition to retirement, but the effect does not last years after transition to retirement.

A9. We agree with the review that this would add to the interpretation of our results, but unfortunately this information is not available. We have mentioned the following in our limitation section: “all participants who retired in a time frame of 4 years were included in this study. There was no information available on the exact time of retirement, while this could have influenced behavioural adjustments.” See lines 278-280, marked copy.

Q10. The authors should tone down the description of the study strengths in the Discussion. First, it is not evident that “Our methods are unique in studying a large variety of behaviours related to both sedentary behaviour and physical activity in different domains” since the methodology is based on self-reported questions. Second, based on the reported results it is not clear how this conclusion can be drawn: “This study adds with valuable knowledge for public health researchers and policy makers that of all sedentary behaviour and physical activity domains, leisure time exercise seems to have the greatest potential in pre-retirement interventions that aim to facilitate healthy aging.” Please see earlier comment for conclusions in the Abstract section and modify accordingly.

A10. We updated these two strengths to:

- The used instrument (PAQ questionnaire) takes into account both physical activity and sedentary behaviour in different domains (e.g. at work, during leisure time)
- This study adds with valuable knowledge for public health researchers and policy makers and indicates that interventions should target those with more unfavourable pre-retirement physical activity and sedentary behaviours pre-retirement, and those interventions focusing on exercise might have greatest potential.

VERSION 2 – REVIEW

REVIEWER	Jean-Michel Oppert Sorbonne university, Paris, France
REVIEW RETURNED	06-Jun-2020
GENERAL COMMENTS	Authors have revised their manuscript in line with comments and suggestions made by this reviewer on the original version. A small remark is: Paragraph Strengths and limitations, lines 278: you

	mention limitations here, but the previous sentence already discuss limitations; adapt the sentence accordingly.
--	--

REVIEWER	Sari Stenholm University of Turku, Finland
-----------------	---

REVIEW RETURNED	17-May-2020
-------------

GENERAL COMMENTS	The authors have addressed well my comments and questions, and the manuscript is much improved. I have no further comments.
---

\